# Dissolved organic matter and sulfide enhance the CH$_4$ consumption of a psychrophilic lake methanotroph, *Methylobacter* sp. S3L5C

Antti Juhani Rissanen,[1,2] Rahul Mangayil,[3] Alexander T. Tveit,[4] Susanna T. Maanoja,[1] Ramita Khanongnuch[1,5]

**ABSTRACT** Gammaproteobacterial methanotrophic bacteria (gMOB) are dominant methanotrophs in the water column of oxygen-stratified boreal and subarctic lakes and ponds. (Meta)genomic data suggest that, besides methane (CH$_4$), gMOB potentially use dissolved organic matter (DOM) and reduced sulfur compounds (e.g., sulfide) as electron sources. To study the DOM and sulfide metabolism of lake gMOB, we subjected a psychrophilic lake water strain, *Methylobacter* sp. S3L5C, first to different sulfide levels (Na$_2$S, 0–5 mM) to test the toxicity, and subsequently, to freshwater DOM (60 mg L$^{-1}$) either alone or with sulfide (0.05 mM) at 1 and 20% CH$_4$ levels. The growth, CH$_4$ and O$_2$ consumption, CO$_2$ production, and mRNA expression patterns of S3L5C were analyzed. Sulfide concentrations of 0–0.5 mM had no effect, while 1 and 5 mM concentrations inhibited the strain's growth. At 20% CH$_4$, DOM addition enhanced CH$_4$ consumption, CO$_2$ production, and growth of S3L5C, while the addition of sulfide+DOM led to further increases in these variables. The addition of sulfide+DOM enhanced CH$_4$ consumption even at 1% CH$_4$. The effect of DOM on the S3L5C's metabolism was accompanied by enhanced expression of the *cyc2* gene, which has been suggested to mediate the extracellular electron transfer from DOM. Furthermore, the addition of sulfide+DOM enhanced the expression of the *sqr* and *soxB* genes encoding dissimilatory sulfide and thiosulfate oxidation, respectively. Together with previous metagenomic data, these results suggest that the usage of DOM and reduced sulfur compounds as electron sources is a trait that enhances methanotrophy among gMOB of boreal and subarctic lakes and ponds.

**IMPORTANCE** Gammaproteobacterial methanotrophic bacteria (gMOB) are crucial mitigators of methane emissions of many ecosystems, like boreal and subarctic lakes and ponds. Metagenomic data suggest that besides using methane, gMOB have genetic potential to use dissolved organic matter (DOM) and sulfide, typically present in lakes and ponds, as electron donors. To test the effect of DOM and sulfide on the methane metabolism of gMOB of oxygen-stratified boreal lakes, we subjected our recently isolated lake gMOB strain, *Methylobacter* sp. S3L5C, to additions of freshwater DOM and sulfide. We show that DOM and sulfide enhance methane consumption and growth of S3L5C. Furthermore, the expression of genes mediating the electron transfer from DOM and sulfide is enhanced. Our results suggest that the usage of DOM and reduced sulfur compounds as electron sources is a trait that enhances methanotrophy among gMOB and adds significantly to the growing body of literature highlighting the enormous metabolic versatility of gMOB.

**KEYWORDS** methanotroph, methane oxidation, *Methylobacter*, lake, pond, greenhouse gas, extracellular electron transfer, dissolved organic matter, sulfide

**Peer Reviewer** Jakob Zopfi, University of Basel, Basel, Switzerland

Address correspondence to Antti Juhani Rissanen, antti.rissanen@tuni.fi.

The authors declare no conflict of interest.

See the funding table on p. 6.

Gammaproteobacterial methanotrophic bacteria (gMOB) are key organisms controlling methane ($CH_4$) fluxes in the water column of oxygen-stratified lakes and ponds (1, 2). These waterbodies are subjected to regional browning due to increasing loads of dissolved organic matter (DOM) and iron (3–5). Browning-induced decrease in light penetration has been suggested to enhance the lake water column $CH_4$ oxidation indirectly through the alleviation of light inhibition of methanotrophy (5). Furthermore, recent metagenomic data show that the genomes of gMOB of DOM-rich lakes and ponds encode extracellular electron transfer (EET), suggesting a potential role of DOM in EET of gMOB, and hence, DOM potentially also directly affects their $CH_4$ metabolism (6). A gMOB strain, *Methylomonas* sp. LW13, was shown to use a model DOM compound (antraquinone-2,6-disulfonate) as an electron acceptor (7), while the role of DOM as an electron donor for gMOB has not been explored.

Oxidation of reduced sulfur compounds (e.g., sulfides) was recently demonstrated with a verrucomicrobial and an alphaproteobacterial methanotroph strain, as was the genetic potential among gMOB to oxidize sulfur compounds (8, 9). As the vertical distribution patterns of sulfide and gMOB partially overlap in the oxygen-stratified waterbodies (10), it can be hypothesized that gMOB use sulfide as an extra electron donor.

Here we investigated the effect of DOM and sulfide on the metabolism of a lake gMOB, the psychrophilic *Methylobacter* sp. S3L5C, recently isolated from the water column of a boreal oxygen-stratified lake (11). To account for sulfide toxicity, previously shown for gMOB species *Methylococcus capsulatus* and *Methylomicrobium album* (12, 13), we initially tested the effect of varying sulfide concentrations (0, 0.05, 0.1, 0.5, 1, and 5 mM) on the growth of S3L5C. Thereafter, we tested the effect of freshwater DOM (60 mg $L^{-1}$) alone and in combination with sulfide (0.05 mM) on S3L5C growth, $CH_4$ and $O_2$ consumption, $CO_2$ production, and mRNA expression patterns at 1 and 20% $CH_4$ levels. DOM was isolated by reverse osmosis combined with electrodialysis from the Upper Mississippi River (1R110N, International Humic Substances Society, C% = 49.98 and N% = 2.36). The applied concentration corresponds to ~30 mg $L^{-1}$ of dissolved organic carbon, which is in the mid-range detected for boreal lakes (10–49 mg $L^{-1}$) (14). All tests were conducted in biological triplicate with abiotic controls (sterile medium without cells). See the detailed methods in the Supplemental materials.

The two highest sulfide concentrations inhibited the growth ($OD_{600}$) of S3L5C, while lower concentrations (0.05–0.5 mM) had no effect (Fig. 1A). Interestingly, there was a statistically nonsignificant tendency for 0.05 mM of sulfide to enhance growth (Fig. 1A). DOM addition enhanced $CH_4$ consumption and $CO_2$ production at 20% $CH_4$ (Fig. 1B) and also had a statistically nonsignificant tendency to enhance $CH_4$ consumption and $CO_2$ production at 1% $CH_4$ (Fig. 1C). Compared to control and DOM treatment, sulfide+DOM addition further increased $CH_4$ consumption and $CO_2$ production at 20% $CH_4$ (Fig. 1B) and also enhanced $CH_4$ consumption compared to control at 1% $CH_4$ (Fig. 1C). DOM addition also had a statistically nonsignificant tendency to enhance growth, while sulfide+DOM addition enhanced growth compared to both control and DOM treatments at 20% $CH_4$ (Fig. 1D). There was also a statistically nonsignificant tendency for DOM+sulfide to enhance growth at 1% $CH_4$ (Fig. 1E). Abiotic controls showed no growth ($OD_{600} = 0$), $CH_4$ consumption or $CO_2$ production (Fig. S1).

The effect of DOM in enhancing the $CH_4$ consumption and growth of S3L5C was accompanied by upregulation of the *cyc2* gene encoding an outer membrane c-type cytochrome (Fig. 2), which typically mediates EET from reduced iron but is speculated to drive also DOM-based EET (6, 15, 16). S3L5C encoded two *cyc2* genes, which differed in their expression patterns between 20% and 1% $CH_4$ (Fig. 2). Furthermore, DOM addition enhanced the expression of $CH_4$ oxidation pathway genes encoding formaldehyde oxidation and assimilation of formaldehyde to ribulose monophosphate (RuMP) cycle as well as nitrogen uptake/assimilation at 1% $CH_4$ (Fig. 2A). In contrast, the expression of genes encoding transcription and translation as well as oxidative phosphorylation (F-type ATPase) was enhanced at 20% $CH_4$ (Fig. 2B).

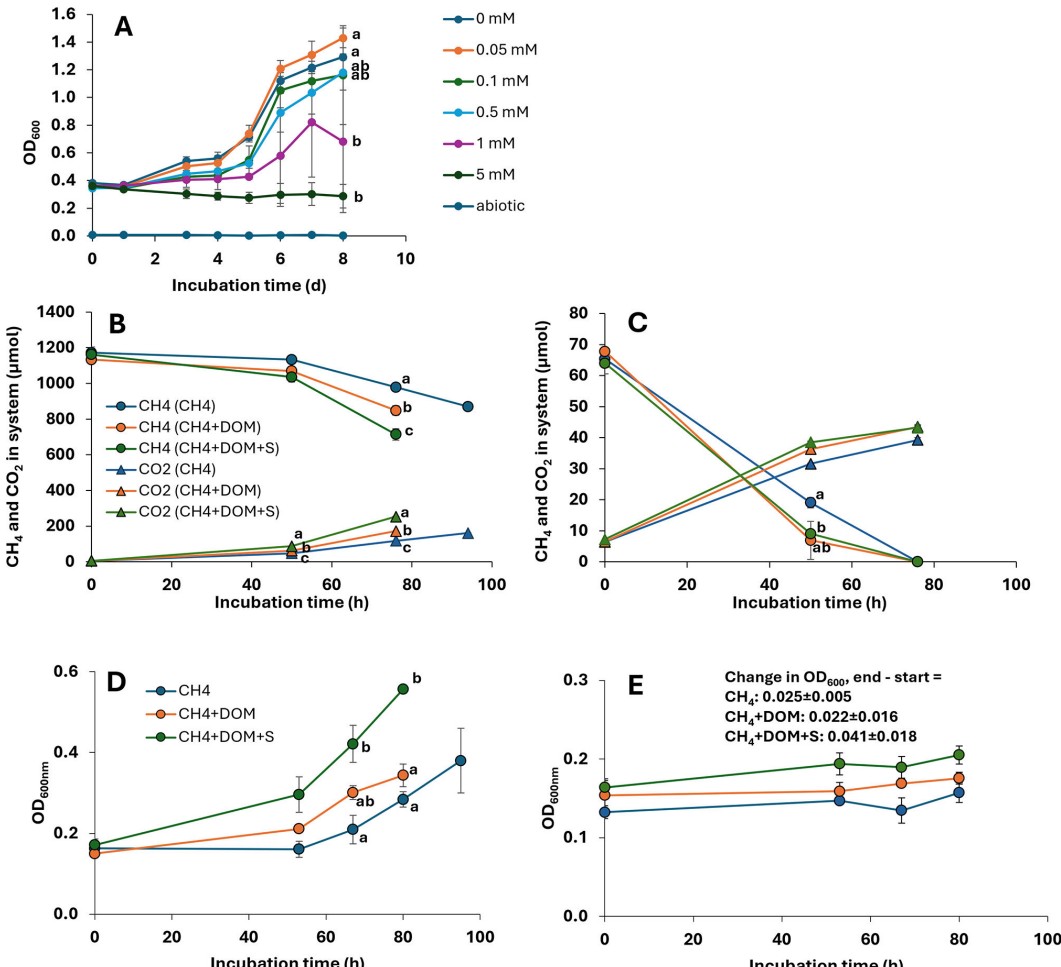

**FIG 1** Growth, $CH_4$ consumption, and $CO_2$ production of *Methylobacter* sp. S3L5C. (A) Biomass growth (optical density, $OD_{600}$) at different sulfide levels in the test for sulfide toxicity. (B, C) $CH_4$ consumption and $CO_2$ production at initial 20% $CH_4$ (B) and initial 1% $CH_4$ (C). (D, E) Biomass growth at initial 20% $CH_4$ (D) and initial 1% $CH_4$ (E). Treatments shown in (B–E) are 1. $CH_4$, 2. $CH_4$+dissolved organic matter (DOM), and 3. $CH_4$+DOM+sulfide. Repeated measures analysis of variance indicated a statistically significant treatment × time interaction ($P < 0.05$) for all other variables except for $OD_{600}$ in the test shown in (E), where only the main effects (time and treatment) were significant ($P < 0.05$). Different letters located close to the data symbols indicate timepoint-specific significant differences (Bonferroni corrected $P < 0.05$) in pairwise tests. For simplicity, the pairwise test results are shown only for the last timepoint for the sulfide toxicity test (A) and for the timepoints having a significant treatment effect for the other tests (for B–D). For (E), we tested, using one-way analysis of variance, that the change in $OD_{600}$ from start (0 hour) to end (80 hours) was not different between treatments ($F = 1.73$; $P > 0.05$). The $CH_4$ and $CO_2$ data for abiotic controls are shown in Fig. S1, while $OD_{600}$ stayed zero throughout the incubations for all abiotic controls.

Besides providing electrons, DOM potentially provides carbon for growth. However, considering that *Methylobacter* spp. and other gMOB have not been observed to grow using multi-carbon compounds (17–19), the growth of S3L5C using DOM as a carbon source is unlikely. Furthermore, transcriptomic analysis suggests that DOM might act as an additional energy source for anabolic processes: glycolysis (Embden-Meyerhof-Parnas [EMP] and Entner–Doudoroff [ED] pathways) and tricarboxylic acid (TCA) cycle gene expression were not enhanced by DOM addition at 20%$CH_4$, suggesting that $CH_4$ remained the primary carbon source (Fig. S2), while glycolysis was slightly upregulated under DOM addition at 1%$CH_4$. Future experiments using stable isotope ($^{13}C/^{12}C$) mass balance or $^{13}C$-labels are needed to conclude whether DOM acts as a carbon source for gMOB. Further studies should also elucidate which DOM functional groups (e.g., quinone, phenolic, carboxyl, hydroxyl, or amino groups) are responsible for the DOM-driven EET. Based on our elemental analyses, the applied DOM also contained iron (but no manganese), with an estimated final concentration in the experiment being ~0.65

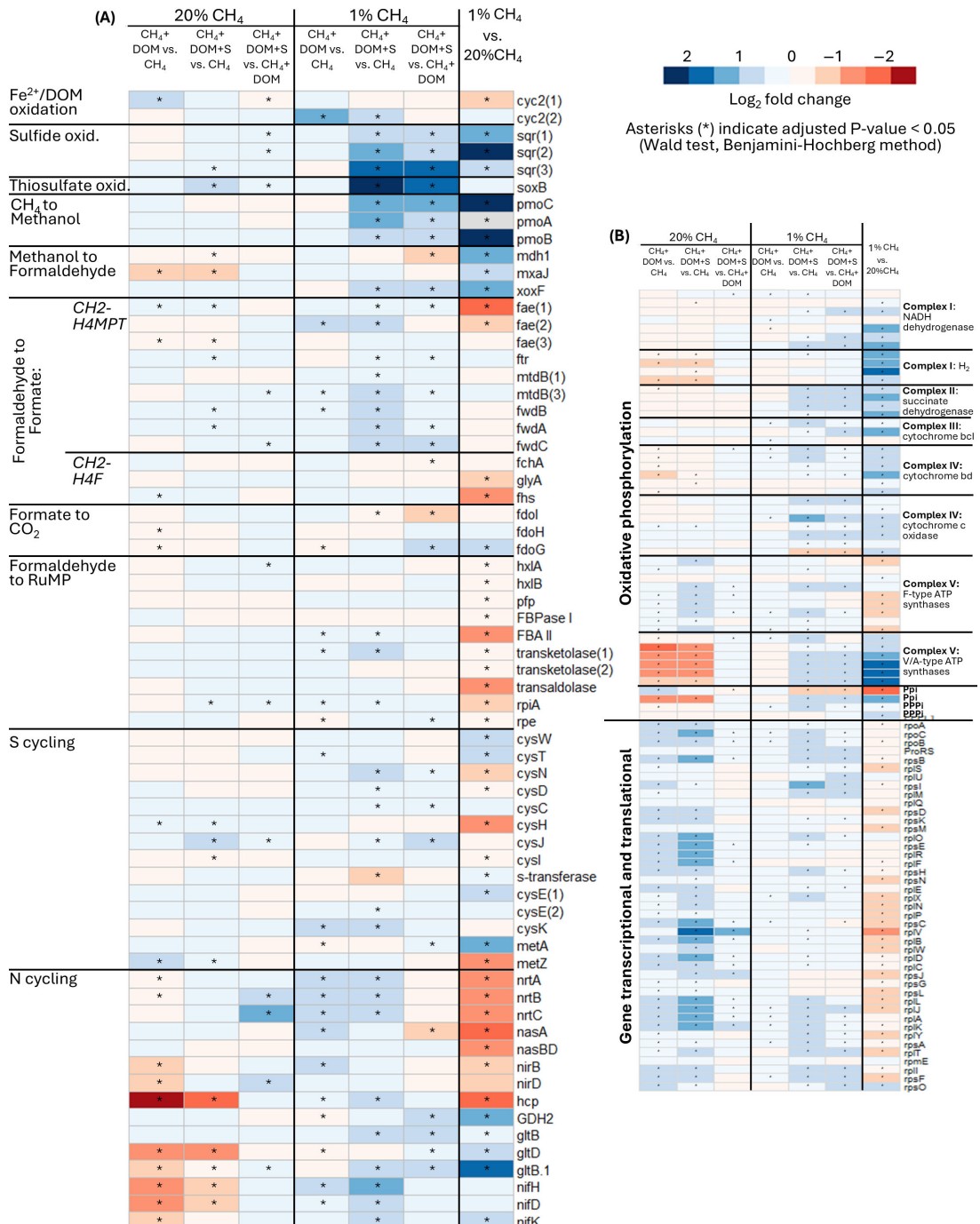

**FIG 2** Heatmap on differential expression of genes of *Methylobacter* sp. S3L5C between treatments. (A) Genes encoding DOM oxidation (extracellular electron transfer from DOM), sulfide oxidation, thiosulfate oxidation, methane oxidation pathway, sulfur cycling (other than sulfide and thiosulfate oxidation), and nitrogen cycling. (B) Genes encoding oxidative phosphorylation, and gene transcription and translation. Compared treatment pairs are 1. $CH_4$+DOM vs $CH_4$, 2. $CH_4$+DOM+S vs $CH_4$, 3. $CH_4$+DOM+S vs $CH_4$+DOM at initial 20% $CH_4$ and 1% $CH_4$, as well as 4. initial 1% $CH_4$ vs initial 20% $CH_4$. Heatmap color shows $log_2$ fold change in gene expression between the treatments, with blue and red colors indicating enhanced or lowered expression of the gene in the first-mentioned treatment compared to the second-mentioned treatment, respectively. Asterisks indicate significant $log_2$ fold change. See full details on gene annotations, normalized counts, and DESeq2 analyses in Table S1.

μmol $L^{-1}$ (see Supplemental materials Section 1.4). The DOM-derived iron increased the $Fe^{2+}$ content of the growth medium approximately by 1/3 (originally in NMS medium 1.8

$\mu$mol L$^{-1}$ iron(II)SO$_4$). The potential significance of this additional iron in contributing to EET needs to be elucidated in future studies.

The effect of sulfide+DOM addition in enhancing CH$_4$ oxidation and growth of S3L5C was accompanied by upregulation of the *sqr* and *soxB* genes encoding the dissimilatory sulfide-oxidizing sulfide-quinone reductase and the SoxB component of the periplasmic thiosulfate-oxidizing Sox enzyme complex, respectively (Fig. 2A) (20, 21). Thiosulfate, present in the liquid medium at 0 hour (0.01 $\pm$ 0.005 mM) but absent at later time points, potentially became available through abiotic sulfide oxidation (22). It was impossible to detect the possible production of sulfate against the high background concentration (5.87 $\pm$ 0.44 mM) originating from the NMS media (Fig. S3). S3L5C encoded three *sqr* genes, which were all upregulated after the addition of sulfide+DOM, yet their upregulation was considerably higher at 1% CH$_4$ (Fig. 2A). Furthermore, sulfide+DOM led to higher expression of genes for methane oxidation from methane (via methanol and formaldehyde) to formate and RuMP cycle, sulfur and nitrogen uptake/assimilation (Fig. 2A), multiple steps in oxidative phosphorylation, as well as translation and transcription at 1% CH$_4$ (Fig. 2B). Correspondingly, at 20% CH$_4$+sulfide+DOM, we saw enhanced expression of genes encoding translation and transcription (Fig. 2B), while the expression of genes for methane oxidation or sulfur and nitrogen uptake/assimilation was not generally upregulated (Fig. 2A). Furthermore, among the genes for oxidative phosphorylation, only ATP synthase was upregulated by sulfide+DOM at 20% CH$_4$, and, interestingly, different ATP synthases were upregulated at 20 and 1% CH$_4$ levels (Fig. 2B).

We suggest that the differences in the expression patterns of *cyc2*, *sqr*, and other genes between 1% and 20% CH$_4$ are potentially attributed to the lower O$_2$ level in 20% CH$_4$ treatment during mRNA sampling (Fig. S4). Correspondingly, the expression of the oxygen-requiring particulate methane monooxygenase (pmoCAB) was considerably higher at 1% than 20% CH$_4$ (Fig. 2A).

Recent metagenomic data (see data set in Olmsted et al. [6], who analyzed theirs and Buck et al.'s [23] metagenomes) indicate that the genetic potential for EET and oxidation of reduced sulfur compounds is widely dispersed among gMOB of boreal and subarctic lakes and ponds (USA, Canada, Sweden, and Finland). This and our observations of increased *cyc2* and *sqr* gene expression in response to DOM and sulfide additions suggest the usage of these compounds by lake gMOB as extra sources of electrons. Hence, besides the indirect effect through alleviation of light inhibition (5), DOM potentially also directly enhances their growth and thus CH$_4$ consumption.

## ACKNOWLEDGMENTS

The authors thank Anne Grethe Hestnes and Mette M. Svenning, The Arctic University of Norway, Tromsø, Norway, for guidance and support in methanotroph cultivation. In addition, Fatemeh Hosseinpour is acknowledged for her help during the laboratory experiments. R.K. acknowledges financial support during manuscript preparation from the PeatlandN2O project, funded by the European Research Council (ERC) under grant agreement No. 101096403.

This study was funded by the Research Council of Finland (former Academy of Finland) (Grant no. 346751 for A.J.R., 353750 for A.J.R. and R.K., and 346983 for R.M.).

A.R.: conceptualization, formal analysis, investigation, resources, writing—original draft, writing—review and editing, supervision, project administration, funding acquisition; R.M.: conceptualization, methodology, writing—review and editing, supervision; A.T.: methodology, supervision, writing—review and editing; S.M.: methodology, formal analysis, investigation, writing—review and editing; R.K.: conceptualization, methodology, formal analysis, investigation, writing—original draft, writing—review and editing, visualization.

## AUTHOR AFFILIATIONS

[1]Faculty of Engineering and Natural Sciences, Tampere University, Tampere, Pirkanmaa, Finland

[2]Natural Resources Institute Finland, Helsinki, Uusimaa, Finland

[3]Department of Bioproducts and Biosystems, School of Chemical Engineering, Aalto University, Espoo, Finland

[4]Department of Arctic and Marine Biology, UiT, The Arctic University of Norway, Tromsø, Troms, Norway

[5]Department of Geography, Institute of Ecology and Earth Sciences, University of Tartu, Tartu, Tartu County, Estonia

## AUTHOR ORCIDs

Antti Juhani Rissanen http://orcid.org/0000-0002-5678-3361

## FUNDING

| Funder | Grant(s) | Author(s) |
|---|---|---|
| Research Council of Finland | 346751, 353750 | Antti Juhani Rissanen |
| Research Council of Finland | 353750 | Ramita Khanongnuch |
| Research Council of Finland | 346983 | Rahul Mangayil |

## AUTHOR CONTRIBUTIONS

Antti Juhani Rissanen, Conceptualization, Formal analysis, Funding acquisition, Investigation, Project administration, Resources, Supervision, Writing – original draft, Writing – review and editing | Rahul Mangayil, Conceptualization, Methodology, Supervision, Writing – review and editing | Alexander T. Tveit, Methodology, Supervision, Writing – review and editing | Susanna T. Maanoja, Formal analysis, Investigation, Methodology, Writing – review and editing | Ramita Khanongnuch, Conceptualization, Formal analysis, Investigation, Methodology, Visualization, Writing – original draft, Writing – review and editing

## DATA AVAILABILITY

Raw transcriptome reads have been deposited in NCBI BioProject database under accession number PRJNA1187248. Gene annotations, normalized counts, and DESeq2 analysis results are provided in Table S1. $CH_4$, $CO_2$, $OD_{600}$, sulfate, and thiosulfate data are provided in Supplementary data set.

## ADDITIONAL FILES

The following material is available online.

### Supplemental Material

**Supplemental materials (Spectrum03133-24-S0001.pdf).** Additional methods and Fig. S1 to S4.

**Supplemental dataset (Spectrum03133-24-S0002.xlsx).** Dataset on CH4, CO2, OD600, and thiosulfate and sulfate.

**Table S1 (Spectrum03133-24-S0003.xlsx).** Full details on gene annotations, normalized counts, and DESeq2 analyses illustrated in Figures 2 and S2.

### Open Peer Review

**PEER REVIEW HISTORY (review-history.pdf).** An accounting of the reviewer comments and feedback.

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
