## [Reviewer comments · Microbiology Spectrum]

Microbiology Spectrum

Dissolved organic matter and sulfide enhance the CH₄ consumption of a psychrophilic lake methanotroph, *Methylobacter* sp. S3L5C

Antti Rissanen, Rahul Mangayil, Alexander Tveit, Susanna Maanoja, and Ramita Khanongnuch

Corresponding Author(s): Antti Rissanen, Tampereen yliopisto - Hervannan kampus

Review Timeline:

Submission Date:	December 4, 2024
Editorial Decision:	February 12, 2025
Revision Received:	April 7, 2025
Accepted:	April 29, 2025

Editor: Sandi Orlic

Reviewer(s): Disclosure of reviewer identity is with reference to reviewer comments included in decision letter(s). The following individuals involved in review of your submission have agreed to reveal their identity: Jakob Zopfi (Reviewer #2)

Transaction Report:

DOI: <https://doi.org/10.1128/spectrum.03133-24>

Re: Spectrum03133-24 (Dissolved organic matter and sulfide enhance the CH₄ consumption of a psychrophilic lake methanotroph, *Methylobacter* sp. S3L5C)

Dear Dr. Antti Juhani Rissanen:

Thank you for the privilege of reviewing your work. Below you will find my comments, instructions from the Spectrum editorial office, and the reviewer comments.

Revision Guidelines

Sincerely,
Sandi Orlic
Editor
Microbiology Spectrum

Reviewer #1 (Comments for the Author):

The manuscript "Dissolved organic matter and sulfide enhance the CH₄ consumption of a psychrophilic lake methanotroph, *Methylobacter* sp. S3L5C" is a careful study with a weakness in the interpretation. Iron(II)/DOM metabolism is linked with the expression of an outer membrane c-type cytochrome (Fig. 2). Throughout the manuscript, the authors consider DOM as electron donating compound. However, the iron content in the DOM used is not

reported and likely unknown. The DOM producer reports only the presence of ash after burning.

The organism originated from boreal lakes that have variable water heights due to rain and evaporation and receive organic matter and ferrous iron from anoxic soil interstitial water during periods of dryness. It would be ideal for the organisms to have a ferrous iron ion oxidation capacity.

Clearly the biomass increase is in my view related to a mixotrophic growth using DOM likely also as carbon source for anabolism. (Do isotope data exist?). But the probability that the electron uptake mechanism is induced by iron ions in the DOM can currently not be excluded and deserves a reflection in the manuscript.

Ideally, an initial experiment on the chemical and biological oxidation of ferrous iron ions with the gammaproteobacterial methane-oxidising bacterium will be included in the study.

Reviewer #2 (Comments for the Author):

see attached file

Review for the manuscript:

“Dissolved organic matter and sulfide enhance the CH₄ consumption of a psychrophilic lake methanotroph, *Methylobacter* sp. S3L5C”

by A.J. Rissanen et al., submitted as an “Observation”-article to Microbiology Spectrum ASM.

Rissanen and colleagues present a concise and well-written manuscript reporting novel experimental data on the potential use of dissolved organic matter (DOM) and sulfide by an aerobic methanotroph. This research builds on recent evidence from genomic and environmental metagenomic data showing that sulfur-cycling and EET genes are widespread in methane-oxidizing bacteria, suggesting that they may also metabolize sulfur compounds, complex dissolved organic matter, or be involved in extracellular electron transfer. Experimental studies testing hypotheses based on (meta-)genomic data, like the submitted one, are highly valuable and welcome. Batch incubation experiments with a psychrophilic *Methylobacter* strain and different combinations and concentrations of DOM and Sulfide have been combined with transcriptome analyses. The methods are up to date, well described and executed. The results are clear and presented in two easily understandable figures. Results show that the presence of lake DOM and sulfide can enhance methane oxidation and growth by *Methylobacter* sp. S3L5C, thus this research adds to the growing body of evidence indicating that aerobic methanotrophs are metabolically much more versatile than previously thought. This has important ecological and biogeochemical implications for our understanding of carbon and sulfur cycling in lacustrine environments. Unfortunately, there is not much discussion and explanation for what is happening in these incubation experiments. I assume, however, that this is the very nature of “observation” articles. Although descriptive, the results are novel and interesting, and are likely to inspire future, more in-depth research on the topic.

Specific comments:

It is stated that nutrients but also sulfate and thiosulfate have been quantified by IC. Do the concentration data indicate sulfate or thiosulfate production in the different experiments with sulfide addition? While one can expect some abiotic sulfide oxidation to thiosulfate in control incubations, the sulfate/thiosulfate increase should be significantly higher in the active culture experiments. These data missing but important to prove e.f. that sulfide may serve as additional electron donor for this methanotroph.

Have you tried to exploit the CH₄, CO₂, and O₂ data any further? E.g. using mass and electron balances in the different assays you could estimate how much of the added sulfide or DOM contributed to O₂ consumption, and potentially, to growth.

Provide information on which controls were used and how they were implemented. (Killed controls? Sterile controls? Without e- donor? ...)

Specific comments to supplementary Methods:

Chapter 1.2

- first line. should be “mid-exponential” and not “mid-logarithmic” growth...
- Was the medium prepared with DI water or MilliQ water?
- Was the H₂S solution prepared from Na₂S·xH₂O and was it neutralized before usage? The solution is quite alkaline, and while neutralization is not necessary, high sulfide additions to a growth medium could lead to pH changes, and potentially to a similar result of stronger growth inhibition at higher sulfide concentrations. Please provide information on the effect of sulfide addition on the pH of the medium.

In supplementary Method:

Chapter 1.3

- Controls are mentioned “in parallel with abiotic controls”. Please show the corresponding data. Either in the main figure or e.g. in the SI 1.

In supplementary Method:

Chapter 1.4

- “anaerobic culture tube” is misleading. All incubations were done under oxic conditions. Were these 25 mL serum bottles? Alternatively, if they are not commonly known serum bottles, provide precise type and supplier.
- Please also provide supplier and type of butyl rubber stoppers as some brands tend to inhibit growth of methanotrophs (or nitrifiers).

Reviewer #1

The manuscript "Dissolved organic matter and sulfide enhance the CH₄ consumption of a psychrophilic lake methanotroph, *Methylobacter* sp. S3L5C" is a careful study with a weakness in the interpretation.

Iron(II)/DOM metabolism is linked with the expression of an outer membrane c-type cytochrome (Fig. 2). Throughout the manuscript, the authors consider DOM as electron donating compound. However, the iron content in the DOM used is not reported and likely unknown. The DOM producer reports only the presence of ash after burning.

The organism originated from boreal lakes that have variable water heights due to rain and evaporation and receive organic matter and ferrous iron from anoxic soil interstitial water during periods of dryness. It would be ideal for the organisms to have a ferrous iron ion oxidation capacity.

Clearly the biomass increase is in my view related to a mixotrophic growth using DOM likely also as carbon source for anabolism. (Do isotope data exist?). But the probability that the electron uptake mechanism is induced by iron ions in the DOM can currently not be excluded and deserves a reflection in the manuscript.

Ideally, an initial experiment on the chemical and biological oxidation of ferrous iron ions with the gammaproteobacterial methane-oxidising bacterium will be included in the study.

Answer: Thank you for the comments. We think that mixotrophic growth is unlikely. Previous papers do not indicate that *Methylobacter* or gammaproteobacterial methanotrophs generally would use multi-carbon sources as their carbon source. However, another major methanotrophic taxa, alphaproteobacterial methanotrophs, do include strains capable of using also multi-carbon compounds as carbon sources. We now add transcriptomic data on glycolysis and TCA cycle gene expression that also points towards CH₄ being the primary carbon source (see below). Our test at 1%CH₄ also did not show any significant DOM-driven growth enhancement. At a later point, the indications from these transcriptomic observations should be tested with carbon isotope mass balance or isotope labeling to conclude whether or not DOM acts as a carbon source for the strain, but this is beyond the scope of the current manuscript.

We did additional characterization of the applied DOM by ICP-MS, and indeed found out that it contains iron (but not another potential electron donating metal, manganese) with estimated final concentration in the experiment being 0.65 μ M. See the ICP-MS analysis description and results in Supplementary Methods Chapter 1.4. In the revised text, we now suggest that future experiments should elucidate which of the DOM components, e.g., quinone, phenolic, carboxyl, hydroxyl or amino functional groups, or even iron, are responsible for the DOM-driven EET.

Besides adding ICP-MS description and results in Supplementary Methods Chapter 1.4, we have now amended our main text to accommodate all the reviewer comments:

At lines 123-139, we now write that "Besides providing electrons, DOM potentially provides carbon for growth. However, considering that *Methylobacter* spp. and other gMOB have not

been observed to grow using multi-carbon compounds [17–19], growth of S3L5C using DOM as a carbon source is unlikely. Furthermore, transcriptomic analysis suggests that DOM might act as an additional energy source for anabolic processes: Glycolysis (EMP and ED pathways) and TCA cycle gene expression were not enhanced by DOM addition at 20%CH₄, suggesting that CH₄ remained the primary carbon source (Supplementary Fig. S2), while glycolysis was slightly upregulated under DOM addition at 1%CH₄. Future experiments using stable isotope (¹³C/¹²C) mass balance or ¹³C - labels are needed to conclude whether DOM acts as a carbon source for gMOB. Further studies should also elucidate which DOM functional groups (e.g., quinone, phenolic, carboxyl, hydroxyl or amino groups) are responsible for the DOM-driven EET. Based on our elemental analyses, the applied DOM contained also iron (but no manganese) with estimated final concentration in the experiment being ~0.65 μmol L⁻¹ (see Supplementary data Section 1.4). The DOM-derived iron increased the Fe²⁺ content of the growth medium approximately by 1/3 (originally in NMS medium 1.8 μmol L⁻¹ iron(II)SO₄). The potential significance of this additional iron in contributing to EET needs to be elucidated in future studies. “

Figure S2. Differential expression of genes involving Embden-Meyerhof-Parnas (EMP) and Entner–Doudoroff pathways and tricarboxylic acid (TCA) cycle, compared across different

treatments: 1. CH₄ + DOM vs. CH₄, 2. CH₄+DOM+S vs. CH₄, 3. CH₄ + DOM + S vs. CH₄ + DOM at initial 20% CH₄ and 1% CH₄, as well as 4. initial 1% CH₄ vs. initial 20% CH₄.

Reviewer #2

Review for the manuscript:

“Dissolved organic matter and sulfide enhance the CH₄ consumption of a psychrophilic lake methanotroph, *Methylobacter* sp. S3L5C”

by A.J. Rissanen et al., submitted as an “Observation”-article to Microbiology Spectrum ASM.

Rissanen and colleagues present a concise and well-written manuscript reporting novel experimental data on the potential use of dissolved organic matter (DOM) and sulfide by an aerobic methanotroph. This research builds on recent evidence from genomic and environmental metagenomic data showing that sulfur-cycling and EET genes are widespread in methane-oxidizing bacteria, suggesting that they may also metabolize sulfur compounds, complex dissolved organic matter, or be involved in extracellular electron transfer. Experimental studies testing hypotheses based on (meta-)genomic data, like the submitted one, are highly valuable and welcome. Batch incubation experiments with a psychrophilic *Methylobacter* strain and different combinations and concentrations of DOM and Sulfide

have been combined with transcriptome analyses. The methods are up to date, well described and executed. The results are clear and presented in two easily understandable figures. Results show that the presence of lake DOM and sulfide can enhance methane oxidation and growth by *Methylobacter* sp. S3L5C, thus this research adds to the growing body of evidence indicating that aerobic methanotrophs are metabolically much more versatile than previously thought. This has important ecological and biogeochemical implications for our understanding of carbon and sulfur cycling in lacustrine environments. Unfortunately, there is not much discussion and explanation for what is happening in these incubation experiments. I assume, however, that this is the very nature of “observation” articles. Although descriptive, the results are novel and interesting, and are likely to inspire future, more in-depth research on the topic.

Answer: Thank you for your comments.

Specific comments:

It is stated that nutrients but also sulfate and thiosulfate have been quantified by IC. Do the concentration data indicate sulfate or thiosulfate production in the different experiments with sulfide addition? While one can expect some abiotic sulfide oxidation to thiosulfate in control incubations, the sulfate/thiosulfate increase should be significantly higher in the active culture experiments. These data missing but important to prove e.f. that sulfide may serve as additional electron donor for this methanotroph.

Answer: Thiosulfate was quantified by ion chromatography (IC) and detected only on day 0 at a concentration of around 0.01 mM (see figure below). Thiosulfate was not detected on subsequent days.

Steady profiles of sulfate concentrations were observed across all experiments (see figure below). However, as we did not measure sulfide, and as the sulfate background in the culture media was high compared to expected changes arising from oxidation of the added sulfide, we cannot use these data to unravel the role of sulfide as an electron donor. Consequently, the changes in sulfate concentrations were not obvious and difficult to detect as initial sulfide used in this study was quite low (0.05 mM in the DOM study).

The absence of thiosulfate after first day, however, hints towards its consumption by the bacteria.

We have now added sulfate data in Supplementary Fig. S3.

The text was also amended as follows (lines 143-147): “Thiosulfate, present in the liquid medium at 0 h (0.01 ± 0.005 mM) but absent at later time points, potentially became available through abiotic sulfide oxidation [22]. It was impossible to detect the possible production of sulfate against the high background concentration (5.87 ± 0.44 mM) originating from the NMS media (Supplementary Fig S3)”

Furthermore, as we do not use nitrate data in the manuscript, we decided to remove its methodology from the Supplemental methods.

Have you tried to exploit the CH₄, CO₂, and O₂ data any further? E.g. using mass and electron balances in the different assays you could estimate how much of the added sulfide or DOM contributed to O₂ consumption, and potentially, to growth.

Answer: No, we have not, and unfortunately, we cannot. As written in the Fig S4 caption, for O₂, there was only one bottle measured per each treatment. In addition, the sensor had a measurement range of 0-2 mg/L, hence, the O₂ consumption change during the first time points could not be measured. The sensors were originally bought for another, hypoxic experiment but we decided to employ them also in this test to get an indication of the differences in O₂ at the beginning and end of the experiment. O₂ was also not measured in treatments with CH₄ and DOM without sulfide. Hence, the O₂ data is too sparse/inadequate to be used in mass/electron balance. However, this is a good point, and we will take it into account for follow-up work.

Provide information on which controls were used and how they were implemented. (Killed controls? Sterile controls? Without e- donor? ...)

Answer: The controls in this study were abiotic controls (sterile nutrient medium without cells).

For clarity, we have modified text as follows:

At lines 97-98 in main text: "All tests were conducted in biological triplicate with abiotic controls (sterile medium without cells)."

Also at Chapter 1.3 in Supplementary methods: "All tests were conducted in biological triplicate and in parallel with abiotic controls (sterile medium without cells)."

We have included abiotic control data in Figure S1 for CH₄ and CO₂, Figure S3 for sulfate, and Figure S4 for O₂. We now also explain in Figure 1 caption (lines 297-299) that “The CH₄ and CO₂ data for abiotic controls is shown in Supplementary Fig. S1, while OD₆₀₀ stayed zero throughout the incubations for all abiotic controls.”

Specific comments to supplementary Methods:

Chapter 1.2

- first line. should be “mid-exponential” and not “mid-logarithmic” growth...

Answer: Corrected

- Was the medium prepared with DI water or MilliQ water?

Answer: DI water was used

- Was the H₂S solution prepared from Na₂S·xH₂O and was it neutralized before usage? The solution is quite alkaline, and while neutralization is not necessary, high sulfide additions to a growth medium could lead to pH changes, and potentially to a similar result of stronger growth inhibition at higher sulfide concentrations. Please provide information on the effect of sulfide addition on the pH of the medium.

Answer: In different sulfide concentration tests, pH was monitored on day 0. The pH of the test at 5mM sulfide concentrations was slightly higher than others (see figure below). However, all these pH values are in an optimal range for S3L5C growth (6.0-7.3, see <https://www.nature.com/articles/s43705-022-00172-x/tables/1>).

“ ”

This has now been reported in supplementary methods chapter 1.2:

“The addition of Na₂S was not shown to increase pH of the medium at sulfide concentrations 0-1 mM, i.e., pH stayed at ~6.8, while a pH increase was observed at 5 mM sulfide, when pH was ~7.3, which, however, is also within the optimal pH range for the growth of S3L5C (6.0-7.3, see Khanongnuch et al. 2022).”

In supplementary Method:

Chapter 1.3

- Controls are mentioned “ in parallel with abiotic controls“. Please show the corresponding data. Either in the main figure or e.g. in the SI 1.

Answer: Data of abiotic controls is shown now in Supplementary Figure S1, S3 and S4.

In supplementary Method:

Chapter 1.4

- “anaerobic culture tube” is misleading. All incubations were done under oxic conditions. Were these 25 mL serum bottles? Alternatively, if they are not commonly known serum bottles, provide precise type and supplier.

Answer: It is actually a cultivation tube and seems to be modified version of screw thread Hungate tubes.

For clarity, we have modified text as suggested:

at Chapter 1.2: “in 25 mL cultivation tubes (18 in diameter × 150 mm in length) equipped with grey rubber stopper (Chromacol 20-B3P, Germany) and aluminum crimp seal.”

- Please also provide supplier and type of butyl rubber stoppers as some brands tend to inhibit growth of methanotrophs (or nitrifiers).

Answer: The grey butyl rubber stoppers were Chromacol 20-B3P, Germany, and we have included this information in the text in Supplementary file (see previous answer). According to Niemann et al., the grey bromo (the one used here) and chloro-butyl rubber stoppers are among the types of rubber stoppers that do not influence methane oxidation rates (<https://aslopubs.onlinelibrary.wiley.com/doi/pdf/10.1002/lom3.10005>).

Re: Spectrum03133-24R1 (Dissolved organic matter and sulfide enhance the CH₄ consumption of a psychrophilic lake methanotroph, *Methylobacter* sp. S3L5C)

Dear Dr. Antti Juhani Rissanen:

Your manuscript has been accepted, and I am forwarding it to the ASM production staff for publication. Your paper will first be checked to make sure all elements meet the technical requirements. ASM staff will contact you if anything needs to be revised before copyediting and production can begin. Otherwise, you will be notified when your proofs are ready to be viewed.

Sincerely,
Sandi Orlic
Editor
Microbiology Spectrum

Reviewer #1 (Comments for the Author):

I thank the authors for their careful revision.

Reviewer #2 (Comments for the Author):

I have no further comments or suggestions.